# Aluminum Supplementation Mediates the Changes in Tea Plant Growth and Metabolism in Response to Calcium Stress

**DOI:** 10.3390/ijms25010530

**Published:** 2023-12-30

**Authors:** Hua Zhang, Yakang Song, Zhenglei Fan, Jianyun Ruan, Jianhui Hu, Qunfeng Zhang

**Affiliations:** 1Tea Research Institute, Chinese Academy of Agricultural Sciences, Hangzhou 310008, China; 2021120430@sdau.edu.cn (H.Z.); jruan@mail.tricaas.com (J.R.); 2College of Horticulture, Qingdao Agricultural University, Qingdao 266109, China; ykks@foxmail.com (Y.S.); ffteazl@stu.qau.edu.cn (Z.F.); hujhtea@163.com (J.H.)

**Keywords:** tea plant, calcium, aluminum, root growth, metabolic profile

## Abstract

Tea plants are more sensitive to variations in calcium concentration compared to other plants, whereas a moderate aluminum concentration facilitates the growth and development of tea plants. Aluminum and calcium show a competitive interaction with respect to the uptake of elements, consequently exerting physiological effects on plants. To further explore these interactions, in this study, we used the solution culture method to treat tea plants with two calcium concentrations (0.8 mM and 5.6 mM) and three aluminum concentrations (0 mM, 0.4 mM, and 1 mM). We then determined the influence of the combined treatments on root growth and quality compound accumulation in the tissues by a combination of phenotype, gene expression, and metabolite analyses. Moderate aluminum supplementation (0.4 mM) alleviated the inhibition of root growth caused by high calcium stress. High calcium stress significantly inhibited the accumulation of most amino acids (e.g., Glutamic acid, Citulline, and Arginine) and organic acids (e.g., a-ketoglutaric acid) in the roots, stems, and leaves, whereas aluminum deficiency significantly increased most amino acids in the roots and leaves (except Serine, Alanine, and Phenylalanine in the roots and Ser in the leaves), with a more than two-fold increase in Arg and Lysine. High calcium stress also induced the accumulation of secondary metabolites such as epigallocatechin gallate and procyanidin in the roots, whereas aluminum supplementation significantly reduced the contents of flavonol glycosides such as quercetin, rutin, myricitrin, and kaempferitrin, as well as caffeine, regardless of calcium concentration. Aluminum supplementation reversed some of the changes in the contents of leaf metabolites induced by calcium stress (e.g., 4-dihydroquercetin, apigenin C-pentoside, phenethylamine, and caffeine). Overall, calcium stress caused severe growth inhibition and metabolic disorders in tea plants, which could be reversed by aluminum supplementation, particularly in maintaining the root tips and the accumulation of secondary metabolites. These results provide a theoretical basis for improving calcium-aluminum nutrient management to promote tea plant growth and quality.

## 1. Introduction

Calcium is an essential nutrient for plant growth. Based on calcium requirements, plants can be classified into calcareous soil plants that require more calcium and calcium-susceptible plants that require less calcium [1], such as peat moss, rhododendron, and blueberry [2,3], along with higher H^+^ concentrations for growth [4,5,6]. Since the tea plant has a lower calcium requirement than the average crop, it is generally considered to be a calcium-susceptible plant [7,8]. The healthy growth of tea plants requires a soil exchangeable calcium level below 3 cmol kg^−1^ and saturation below 34% [9,10]. The low efficiency of Ca^2+^ partitioning in calcium-susceptible plants leads to high Ca^2+^ concentrations in the protoplast, which can harm the plant [11,12,13]. The calcium level in tea plants ranges from 1.4 to 5.6 g kg^−1^, and calcium is largely present in mature tissues in the form of ions, salts, or organic compounds. An excessive level of exchangeable calcium in the soil (>5 cmol kg^−1^) is unfavorable to the growth of tea plants, and it did wildly happen in regions with higher latitudes, such as Shandong and Shaanxi Province, China. Under calcium stress, the main root of the tea plant becomes shallow, the absorbing root is scarce, the thick root grows in a spiral, the root skin is black with many “small pimples”, and the skin might even peel off, causing root rot; moreover, the number of new sprouting shoots decreases and the number of pinched leaves increases, resulting in an overall reduction in tea yield [14]. A previous study showed that the inhibition of root growth in tea plants is the main cause of calcium toxicity due to excessive calcium treatments [15]. Although tea plants require less calcium, they still need a certain level of calcium to maintain physiological functions [16]. Importantly, calcium can neutralize organic acids in the tea plant. The metabolite oxalic acid interacts with calcium to form insoluble calcium oxalate crystals in the tea plant, which prevents over-acidification and toxicity while regulating acidity within the plant body, facilitating the normal conversion and transport of assimilated substances [17].

In contrast to calcium, aluminum is a relatively specific element for the tea plant, which can be accumulated in large amounts in the body, reaching up to 1–2% in the old leaves [18]; therefore, the tea plant is considered a typical poly-aluminum plant [19]. According to the concentration, aluminum can be toxic or promote the growth and development of tea plants. Adequate addition of aluminum (10–20 mg L^−1^) can promote the uptake and accumulation of aluminum in tea plants while reducing calcium uptake in the roots but not in the stems and leaves [20]. Another study showed that an aluminum addition of 15 μM could promote new shoot and root growth; however, as the aluminum content of the mature leaf and stem tissues increased, the high concentration of aluminum caused disruption of the oxidative system, leading to a range of biochemical changes [21]. Specifically, new growth of the tea plant reached a maximum with a 50 μM aluminum concentration, while the root biomass increased by threefold with the supplementation of 300 μM aluminum, and the total root length was positively correlated with the root aluminum concentration (r = 0.98). Aluminum induces and stimulates tea plant growth by coordinating high photosynthetic rates and increased antioxidant defenses, and the large root surface area increases water and nutrient uptake by the plant [22]. Aluminum may exist in the form of complexes when absorbed by the tea plant, thus alleviating its potential toxic effects [19,23]. Aluminum is transported to sites such as the adult leaves via the formation of complexes between aluminum and fluorine or phosphorus, among other elements when absorbed into the body of the tea plant. In the tea plant, aluminum predominantly exists in a citrate-bound state in the xylem fluid, which is then transported from the roots to the ground. Hajiboland et al. [22] suggested that aluminum may alleviate iron toxicity in tea plants by reducing iron accumulation and translocation, thereby promoting tea plant growth.

Aluminum inhibits calcium uptake by competing with calcium for binding sites, obstructing calcium ion channels in the protoplasmic membrane, and disrupting calcium homeostasis [24]. Thus, an increased aluminum concentration reduces calcium uptake and accumulation in the tea plant, likely owing to enhanced competition at the common binding sites. By contrast, an increased calcium concentration can reduce aluminum uptake and accumulation in the roots and improve calcium uptake in tea plants. This can be explained by the enhanced competitiveness of calcium ions for aluminum ions at their binding sites in the cell membrane, the reduced activity of aluminum ions and their blocking of calcium ion channels, and the increased entry of calcium ions in the cytosol [20,25]. At present, there are two means applied to reduce plant calcium poisoning: one is to reduce the soil pH, and the other is to use ion antagonism to reduce plant calcium uptake. Aluminum sulfate is often used to improve the growth of tea plants in actual production [26,27].

Thus, considering that aluminum supplementation promotes root growth and affects the expression of some genes [20,21] and aluminum’s mitigating effect on calcium [24], we hypothesized that aluminum supplementation could promote tea plant growth and tea quality under calcium stress. In this study, to reveal the mechanism by which aluminum supplementation mediates the changes in tea plants in response to calcium stress, we monitored the effects of calcium and aluminum on the growth and quality of tea plants according to the effects of ion uptake, metabolite distribution, and gene expression. These findings can provide a theoretical basis for improving the nutritional management of calcium and aluminum in tea plantations.

## 2. Results and Discussion

### 2.1. Effects of Calcium and Aluminum Treatments on the Root Growth of Tea Plants

Calcium is not only involved in building the structural material of cells [28] but also acts as a secondary messenger, regulating the process of plants’ responses to changes in the external environment [29,30,31]. The most active part of the root system is the milky-white absorbing root, which is the site of important life activities such as root elongation, differentiation of mature tissues, and absorption of water and inorganic salts [32]. Absorbent roots decay and renew during plant growth and development, and the root cells of calcium-susceptible plants cannot effectively control the cytoplasmic calcium ion concentration. In this study, compared with normal calcium treatment (0.8 mM), the root color of tea plants was darker under the high calcium stress condition (Figure 1). The shoot biomass of the tea seedlings was significantly lower in the high calcium group than in the low calcium group at the same aluminum concentration. The shoot biomass of tea plants under calcium stress was significantly increased by aluminum addition (relative to that measured under aluminum deficiency) but was not significantly different from that of normal calcium-treated tea plants (Figure 2). The content of aluminum in the root increased with increasing concentrations of aluminum treatments. Compared to those of the high calcium (5.6 mM) treatment group, the content of aluminum was significantly higher and the content of calcium was significantly lower in the roots under the low calcium concentration (0.8 mM) condition. Moreover, the calcium content in the root decreased with increasing aluminum concentration under the low calcium concentration (0.8 mM) (Figure 2).

The aluminum content in the root was significantly increased with a low calcium concentration (0.8 mM) compared to that of the high-calcium (5.6 mM) treatment group, while the calcium content in the root decreased with an increasing aluminum concentration in all treatment groups. This finding supports an antagonistic interaction between Ca^2+^ and Al^3+^ in the cells of the tea plant. Under high calcium stress, low doses of aluminum were beneficial for tea plants to maintain root growth, and the addition of aluminum reduced the content of calcium ions absorbed by tea plants. Aluminum also had a certain inhibitory effect on calcium uptake in the root system of tea plants, suggesting that aluminum can alleviate the toxic effect of calcium ions in tea plants to a certain extent.

Tea plants are typical aluminum-aggregating plants; this means that aluminum tends to accumulate in large quantities, especially in the old leaves and root system [18,19]. As the aluminum concentration increases, more aluminum is partitioned to the roots and chelates with the organic acids secreted in the root system; thus, aluminum fixation at the inter-root level is an important aspect of aluminum tolerance in tea plants [33]. At the same calcium concentration, the root length, number of root tips, and root surface area of tea plant roots were significantly longer in the 0.4 mM aluminum treatment group than in the aluminum-deficient treatment group, whereas these parameters did significantly differ between the 1 mM aluminum and aluminum-deficient conditions (Figure 3). It is noteworthy that the number of tea plant root tips was significantly reduced under high calcium stress in the aluminum-deficient condition but was not affected by increasing calcium concentrations after aluminum supplementation.

In this experiment, the fresh weight of tea seedlings increased to different degrees after the addition of aluminum, with a more evident promotion effect detected with treatment with a low concentration of aluminum; however, there was no significant difference in the root length, number of root tips, or root surface area between the 1 mM aluminum and aluminum-deficient treatment groups. The addition of aluminum to the 0.8 mM calcium treatment group promoted the growth of the root system and further increased the fresh weight of the tea seedlings, root length, and number of root tips, with a greater promotion effect detected with treatment of 0.4 mM than with 1 mM aluminum.

### 2.2. Differential Metabolites Identified with UPLC-Q-TOF/MS

Using the UPLC-Q-TOF/MS analytical platforms, 1126 (447 negative and 679 positive), 1369 (564 negative and 805 positive), and 1278 (569 negative and 709 positive) metabolites were detected from the roots, stems, and leaves, respectively, under six combined treatments of 0.8 and 5.6 mM calcium and 0, 0.4, and 1 mM aluminum, respectively; among these, a total of 580 compounds co-appeared in the three tissues (Figure 4a). Based on the detected metabolites, the three tissues were well separated in the PCA plot (Figure 4b). The first two principal components of the PCA model explained 42.3% of the total variance in the roots, 36.4% of the total variance in the stems, and 30.1% of the total variance in the leaves (Figure 4c–e).

Based on the standard of VIP > 1 and *p* < 0.05 in the PLS-DA model, 284, 179, and 304 differential metabolites were screened at the tea plant roots, stems, and leaves, respectively, under the two different calcium concentration treatments. Only three differential metabolites were common to all three tissues, namely theaflavin monogallate, fructose, and quinic acid (Figure 5a).

We further compared the metabolites identified in the tea plant roots, stems, and new shoots in response to the three aluminum treatments with calcium concentrations of 0.8 mM and 5.6 mM, respectively. A higher number of metabolites was detected in response to the aluminum-deficient treatment of tea plants than in response to the high concentration of aluminum (1 mM), using the normal aluminum treatment (0.4 mM) as the control (Figure 5b–d). The typical differential metabolites included amino acid derivatives, carbohydrates, organic acids (Figure 5e), and secondary metabolites (Figure 6). Among these, flavonoids were the most abundant, including catechins and their derivatives, flavonoids and flavonols, phenols and their derivatives, and anthocyanins.

Tea plants are a calcium-anaerobic plant type, and Ca^2+^ and Al^3+^ are antagonistic in the plant cells. The inefficiency of Ca^2+^ in cell division by calcifying plants leads to an excessive protoplasmic Ca^2+^ concentration, which causes damage to the plant. When calcareous plants are subjected to excessive calcium stress, the root is the first tissue to be affected [34,35]. High calcium stress significantly inhibited the accumulation of most amino acids (e.g., Glu (Glutamic acid), Cit (Cittulline), and Arg (Arginine); Figure 5) and organic acids (e.g., a-ketoglutaric acid) in the tea plant roots, stems, and leaves compared to the corresponding levels detected in the normal calcium treatment group. In the tea plant roots and leaves, aluminum deficiency induced a significant increase in most amino acids (except Ser (Serine), Ala (Alanine), and Phe (Phenylalanine) in the roots and Ser in the leaves), including a more than 2-fold decrease in Arg and Lys (Lysine), whereas aluminum supplementation resulted in a decrease in amino acids. Consequently, many amino acids (e.g., Asp, Phe, Cit) and organic acids (e.g., a-ketoglutaric acid) had the lowest levels under the condition of high calcium and excess aluminum and had the highest levels in the low-calcium and aluminum-deficient condition.

Calcium and aluminum also induced changes in the carbohydrate content in various parts of the tea plant (Figure 5), although the pattern was relatively complex. Aluminum significantly inhibited fucose, fructose, and lyxose accumulation in the leaves but induced sucrose and mannose-6-phosphate accumulation. At the same time, high calcium stress increased the sucrose content in the leaves but inhibited fucose accumulation in the stems.

By contrast, high calcium stress induced the accumulation of secondary metabolites in the roots, such as epigallocatechin gallate and procyanidin, whereas some of these compounds were significantly reduced by the supplementation of aluminum, including quercetin (Figure 6). In addition, leaf secondary metabolites were affected by both calcium and aluminum supplementation, with the latter treatment significantly reducing flavonol glycosides such as quercetin, rutin, myricitrin, and kaempferitrin, as well as caffeine under both calcium concentrations. It is noteworthy that aluminum supplementation reversed some of the changes in leaf metabolite contents caused by high calcium stress. For example, the contents of dihydroxybenzaldehyde, 4-hydroxybenzaldehyde, dihydroquercetin, and apigenin C-pentoside in the leaves decreased under high calcium stress but increased after aluminum supplementation. Conversely, the contents of phenethylamine, isorhamnetin-3-O-neohespeidoside, and caffeine were increased by high calcium stress but decreased by aluminum supplementation.

Morita et al. [19,23], found that after absorption in the plant body, aluminum is transported to the adult plant leaves in the form of complexes with fluorine or phosphorus. In addition, the aluminum in the xylem sap transported from the roots to the aboveground parts of the tea plant predominantly exists in a citrate-bound state. Thus, these aluminum complexes may combine with cationic calcium to inhibit the toxic effect of high calcium stress. Indeed, the present study showed that high calcium stress significantly reduced the number of tea plant root tips under a state of aluminum deficiency, whereas the root tips were not affected by an increasing calcium concentration under aluminum supplementation. In addition, aluminum supplementation could reverse some of the changes in the contents of leaf metabolites induced by calcium stress, including 4-dihydroquercetin, apigenin C-pentoside, phenethylamine, and caffeine (Appendix A).

### 2.3. Gene Expression Analysis

*XTH* encodes xyloglucan endoglycosyltransferase/hydrolase, which plays an important regulatory role in the growth of plant lateral roots, along with the *NAC* family (*NAM*, *ATAF*, *CUC1*) of transcription factors. *XTH* is also associated with root and flower development, fruit maturation, and lignification formation. Under aluminum-deficient conditions, calcium stress induced the expression of the tea plant root cell elongation-associated genes *XET1* (xyloglucan endotransglucosylase 1) and *XTH* (associated with root development), as well as the calmodulin genes *CaM1* (Calmodulin 1) and *CaM2* (Calmodulin 2), but suppressed the expression of the stress-responsive gene *CDPK12* (Calcium-dependent protein kinase 12) and the transcription factor *NAC1* (Figure 7). After aluminum supplementation in tea plants, the expression levels of the root growth-associated genes *XET1* and *XTH* significantly increased; however, excess aluminum significantly inhibited the expression of these two genes, especially under calcium stress. Moreover, the expression level of *CDPK12* was significantly increased after treatment with 0.4 mM aluminum compared to that in the aluminum-deficient group under a fixed concentration of 5.6 mM calcium, reaching a similar level to that detected with a normal calcium level and aluminum deficiency. This suggests that a moderate amount of aluminum is also important for maintaining *CDPK12* expression under calcium stress. Therefore, the gene expression analysis results were in line with the phenotypic responses to aluminum and calcium treatments.

## 3. Materials and Methods

### 3.1. Plant Materials

One-year-old tea seedlings (Camellia sinensis cv. Longjing 43) were used for hydroponic experiments. The seedlings were cultivated according to the methods described by Liang et al. [36] The experiment was a two-factor test by Ca and Al, where Ca had two levels of 0.8 mmol and 5.6 mmol, and Al had three levels of 0 mmol, 0.4 mmol, and 1 mmol, for a total of six treatments. The calcium and aluminum treatments were provided with CaSO_4_ and AlCl_3_, respectively, and the experiment was carried out for 11 weeks (Figure 8). At the end of the experiment, shoots, expanded leaves, stems, and root samples were randomly selected and collected, washed using distilled deionized water, frozen with liquid nitrogen, and then stored in an ultra-freezer at −80 °C. The samples from each treatment were analyzed with three biological replicates. 4.2. Root morphology and element determination

The root morphology and parameters of the tea seedlings were scanned and analyzed by a multi-functional root-scanning analyzer (Perfection Flatbed Scanner V900; Epson, Suwa, Japan) coppered with WinRHIZO 2012 software (Regent Instruments, Quebec, QC, Canada). The content of Ca and Al was analyzed by inductively coupled plasma-atomic emission spectrometry (refer to Liang et al.) [37].

### 3.2. Metabolomics Analyses

Metabolites were identified and quantified by gas chromatography and ultra-high-performance liquid chromatography-quadrupole-time-of-flight mass spectrometry according to modified versions of previously reported methods [36]. Specific measurements refer to Liang et al.

### 3.3. Quantitative Real-Time Polymerase Chain Reaction (qRT-PCR) Analysis

The kits and instruments required for determination and primer pairs used for qRT-PCR were referred to Liang et al. [37], and GAPDH was used as the reference gene [36]. Relative transcript levels were calculated against those of the internal control (GAPDH) using the formula 2^−ΔΔCt^.

### 3.4. Statistical Analysis

Unsupervised principal component analysis (PCA) was performed using SIMCA-P version 13.0 software (Umetrics, Umea, Sweden) to obtain a general overview. The supervised projection-to-latent structure discriminant analysis (PLS-DA) method was used to extract maximum information. The combination of p(corr) and variable importance in the projection (VIP) values from the PLS-DA was used as the coefficients for metabolite selection (VIP > 1.0 and |p(corr)| > 0.5). Data were statistically evaluated by a Student’s *t*-test or one-way analysis of variance as appropriate in SPSS version 15.0 (SPSS Inc., Chicago, IL, USA); *p* < 0.05 indicated a statistically significant difference.

## 4. Conclusions

Calcium stress causes severe growth inhibition and metabolic disorders in tea plants, whereas an appropriate dose of aluminum facilitates root growth and promotes the accumulation of valuable metabolites. The status of the aluminum supply clearly protected against negative changes in root growth and the accumulation of leaf metabolites in tea plants in response to calcium stress. However, the ability of aluminum to alleviate the inhibitory effects of calcium stress on plant growth was limited, which was only reflected in the improvement of root tip number and the accumulation of some secondary metabolites. Overall, this comprehensive analysis of the effects of calcium-aluminum interactions on tea plant growth and quality can provide a theoretical basis for improving calcium-aluminum nutrient management in tea plantations.

## Figures and Tables

**Figure 1 ijms-25-00530-f001:**
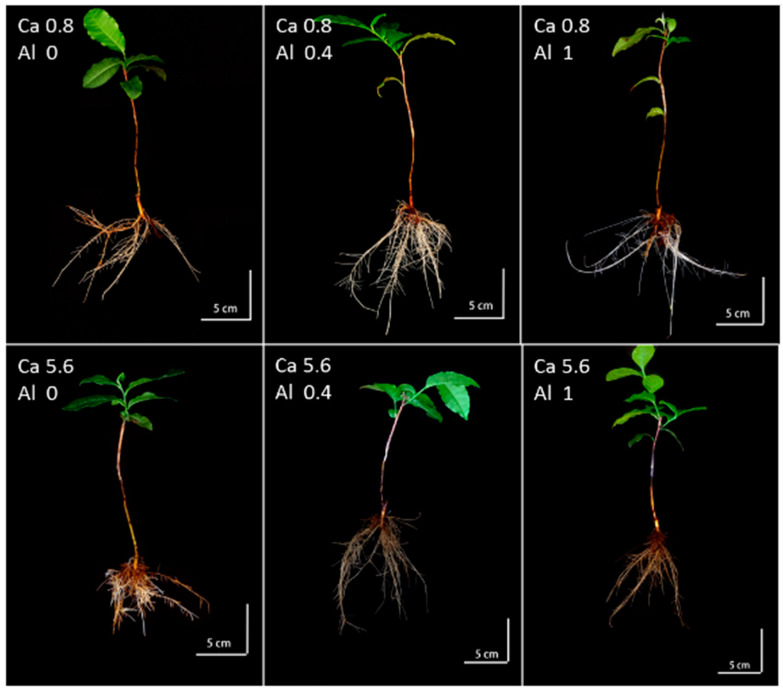
The different phenotypes of single plantlets treated with different calcium and aluminum concentrations.

**Figure 2 ijms-25-00530-f002:**
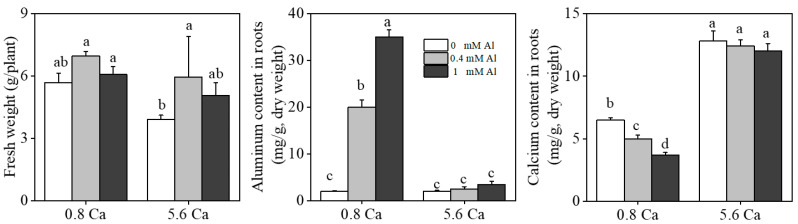
Total biomass of single plantlets treated with different calcium and aluminum concentrations; Ca and Al contents in tea roots after Ca and Al treatments Error bars represent the mean ± standard deviation (*n* = 3). Different letters above the bar indicate significant differences among treatments (*p* < 0.05).

**Figure 3 ijms-25-00530-f003:**
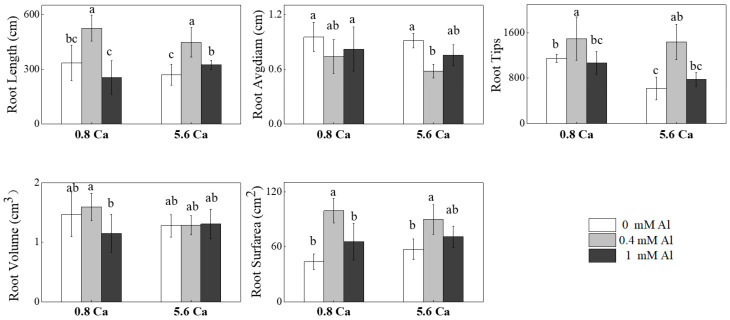
Root Length, Root Avgdiam, Root Tips, Root Volume, and Root Surfarea of roots treated with different calcium and aluminum concentrations. Error bars represent the mean ± standard deviation (*n* = 3). Different letters above the bar indicate significant differences among treatments (*p* < 0.05).

**Figure 4 ijms-25-00530-f004:**
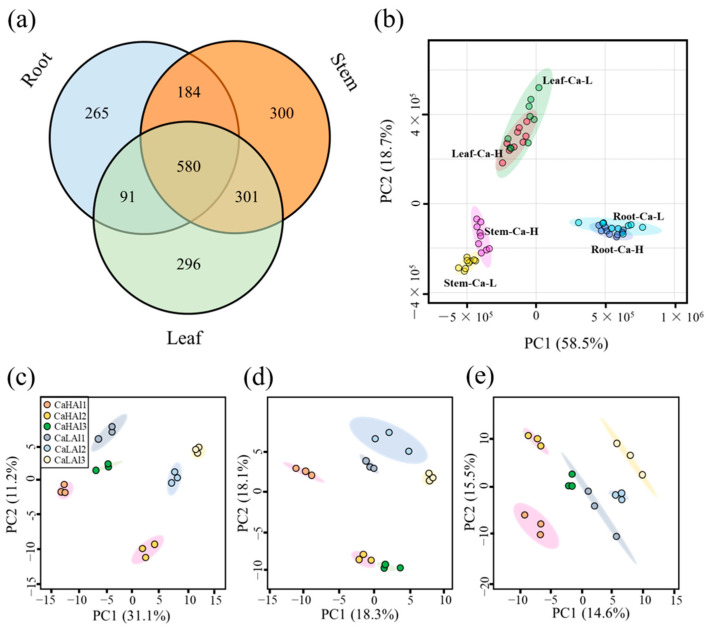
Venn plot (**a**) and PCA score plot (**b**) of characteristic metabolites in various parts of roots, stems, and leaves; PCA score plot of metabolites in various parts of roots (**c**), stems (**d**), and leaves (**e**) under different calcium and aluminum concentrations. Ca-L, 0.8 mmol CalciumL; Ca-H, 5.6 mmol CalciumL; CaLAl1, 0.8 mmol Calcium, 0 mmol Aluminum; LCaLAl2, 0.8 mmol Calcium, 0.4 mmol Aluminum; CaLAl3, 0.8 mmol Calcium, 1 mmol Aluminum; CaHAl1, 5.6 mmol Calcium, 0 mmol Aluminum; CaHAl2, 5.6 mmol Calcium, 0.4 mmol Aluminum; CaHAl3, 5.6 mmol Calcium, 1 mmol Aluminum.

**Figure 5 ijms-25-00530-f005:**
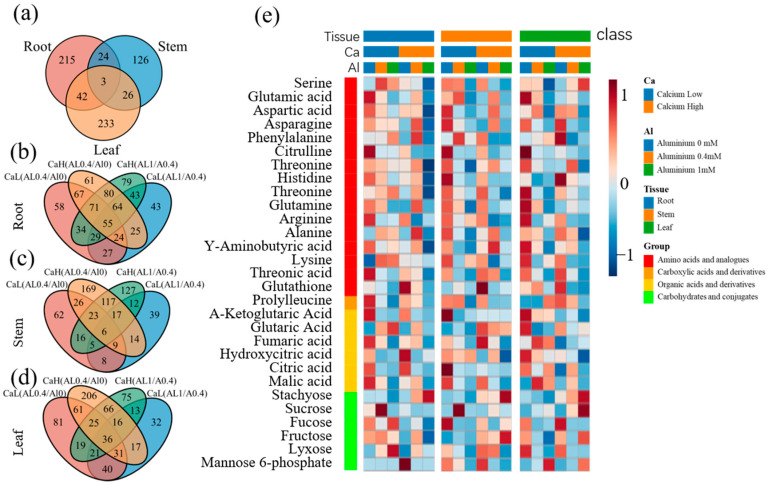
Venn plot of differentially expressed metabolites under different calcium treatments (**a**); Venn plot of differentially expressed metabolites in various parts of roots (**b**), stems (**c**), and leaves (**d**) under different aluminum treatments; Heatmap (**e**) of primary metabolites under different calcium and aluminum concentrations. AlL, 0 mmol Aluminum; AlM, 0.4 mmol Aluminum; AlH, 1 mmol Aluminum; CaL, 0.8 mmol CalciumL; CaH, 5.6 mmol CalciumL; R, roots; S, shoots; L, leaves.

**Figure 6 ijms-25-00530-f006:**
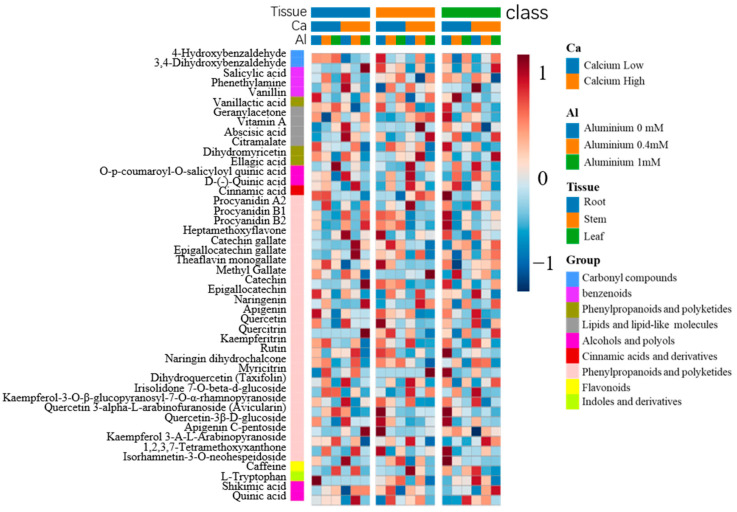
Heatmap of secondary metabolites under different calcium and aluminum concentrations. AlL, 0 mmol Aluminum; AlM, 0.4 mmol Aluminum; AlH, 1 mmol Aluminum; CaL, 0.8 mmol CalciumL; CaH, 5.6 mmol CalciumL; R, roots; S, shoots; L, leaves.

**Figure 7 ijms-25-00530-f007:**
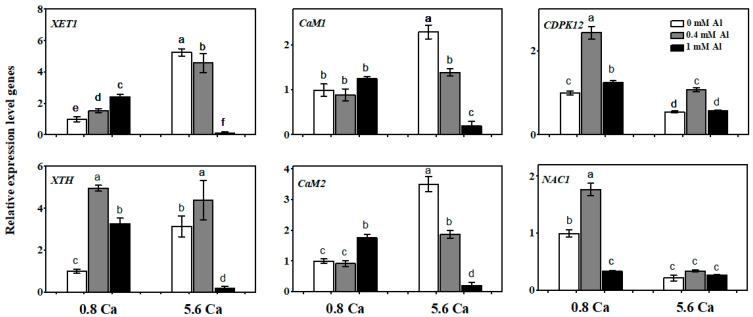
Gene expression in tea roots under Calcium and Aluminum Treatment. *XET1* (xyloglucan endotransglucosylase 1), *XTH* (xyloglucan endoglucanase transglucosylase/hydrolase protein), *CaM1* (Calmodulin 1), *CaM2* (Calmodulin 2), *CDPK12* (Calcium-dependent protein kinase 12), *NAC1* (*NAM, ATAF, CUC1*). Different letters above the bar indicate significant differences among treatments (*p* < 0.05).

**Figure 8 ijms-25-00530-f008:**
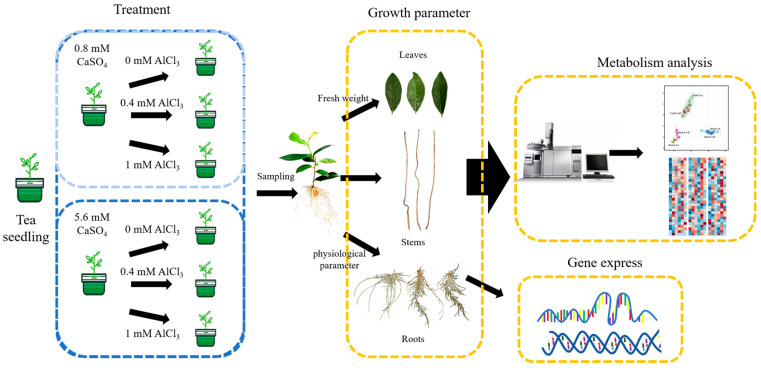
Methodological and experimental graphical abstract.

## Data Availability

Data are contained within the article and Appendix A.

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
