# Peer review of "Aluminum Supplementation Mediates the Changes in Tea Plant Growth and Metabolism in Response to Calcium Stress"

_ijms, 2023, doi:10.3390/ijms25010530_

Round 1
Reviewer 1 Report
Comments and Suggestions for Authors
Dear Editor
The paper aims at the influence of Al supply on tea cultivation under calcium stress. It fits with the research subject of the journal. Three Al concentrations were used in tea culture in a greenhouse under two different Ca concentrations and plant growth, secondary metabolites and gene expression were measured.
It is well known that Al is essential for root growth and development of tea plants which are hyperaccumulators and can tolerate the high Al content in the soil.
Much research has been done on the application of exogenous Ca in order to ameliorate drought conditions or high Al concentrations and other stresses in tea growth. In this paper exogenous Al doses were used in order to alleviate the Ca stress. But how often does it happen?
Even though much work has been done in the present study, more information is needed eg some absolute values of significant metabolites in roots and shoots. Venn plot, PCA and heatmap are useful tools when they correlated with absolute values.
Page 3, lines 114-115: the conclusion is wrong since no significant changes to Al and Ca concentrations occurred under high Ca concentration (Fig 2).
Into the legends of Figs 2 and 3 the authors should add average values of some replications (means of n=?).
I suggest that the manuscript can be accepted for publication after major revision.
Author Response
Reviewer: 1
Comments to the Author:
The paper aims at the influence of Al supply on tea cultivation under calcium stress. It fits with the research subject of the journal. Three Al concentrations were used in tea culture in a greenhouse under two different Ca concentrations and plant growth, secondary metabolites and gene expression were measured. It is well known that Al is essential for root growth and development of tea plants which are hyperaccumulators and can tolerate the high Al content in the soil. Much research has been done on the application of exogenous Ca in order to ameliorate drought conditions or high Al concentrations and other stresses in tea growth.
- In this paper exogenous Al doses were used in order to alleviate the Ca stress. But how often does it happen?
Response: Calcium stress did widely happen in tea plant cultivation, mainly because tea plants are sensitive to calcium levels and are considered as calcium-susceptible plants. Moreover, a high content of calcium can be found in tea garden soils from regions with higher latitudes, such as Shandong and Shaanxi, China. We added related content to the main text (Line 46-47).
- Even though much work has been done in the present study, more information is needed eg some absolute values of significant metabolites in roots and shoots. Venn plot, PCA and heatmap are useful tools when they correlated with absolute values.
Response: We added related results in the tables S1, S2 as suggested. - Page 3, lines 114-115: the conclusion is wrong since no significant changes to Al and Ca concentrations occurred under high Ca concentration (Fig 2).
Response: This sentence be check and modified and the conclusion be revised to “under the low calcium concentration (0.8 mM)”. (Line127-128)
- Into the legends of Figs 2 and 3 the authors should add average values of some replications (means of n=?)
Response: Added as suggested . (Line146-148, 172-174, Figs 2 and 3)
Reviewer 2 Report
Comments and Suggestions for Authors
The authors conducted an interesting analysis measuring root development and changes in shoot metabolites in green tea due to Al supply during Ca stress. Some detailed corrections have been recorded in detail in the attached file. Please find the attached file.

Detail suggested corrections have been recorded in detail in the attached file. Please find the attached file.
Author Response
Reviewer: 2
Comments to the Author:
The authors conducted an interesting analysis measuring root development and changes in shoot metabolites in green tea due to Al supply during Ca stress. Some detailed corrections have been recorded in detail in the attached file. Please find the attached file.
- Please indicate upper and lower cases accurately throughout the whole manuscript.
Response: The whole manuscript be checked throughout and revised according to the reviewers' comments. (Line 38-44, 64, 217-219) - In the picture, the lettering size indicating the significance of the standard deviation is different. Please unify and indicate the p value of the standard deviation.
Response: Figures were checked and modified according to the reviewer's suggestions, and p-values be added.
- What does the word mean of Length, avgdiam, tips and surfarea. And please check the units.
Response: The means of those words be explain in the Figure legend. The text was checked and revised according to the reviewers' comments. (Line 170,171)
- If this is expressed as gene name and species name, it should be expressed in italics. And when abbreviations and full names occur, use ( )
Response: Revised as suggested. (Line 261-176, 343-346)
5.What is the basis for this sentence? Please clarify the basis. This is a very difficult expression to understand. (Line 239-241)
Response: Previous studies have shown that excess soil exchangeable calcium is harmful to tea plants (Journal of the Science of Food and Agriculture. 2007. 87, 1505–1516.), while Aluminum inhibits calcium uptake by competing with calcium for binding sites. This point has been addressed in the text (Line 40-49,86-87). so, aluminum can alleviate the toxic effect of calcium ions in tea plants to a certain extent
- Please check the lower case lettering on the reagent labeling. Please check and write clearly with case distinction, including numbers.
Response: We have checked carefully and rewrite the Materials and Methods section according the editor comments, and this section has been deleted.
Reviewer 3 Report
Comments and Suggestions for Authors
The paper is interesting and provides interesting information for a broad audience, but there are some points that must be adressed befor beig suitable for publication.
Introduction: In general aluminum is a problem, specifically in acidic soils which are very common in tropical agriculture. Wouldn't it be easier to acidify the soil that supplementing a toxic like aluminum? Adding aluminum would not be a problem in the long term? Please comment on this in the introduction.
Figure 2: in the right panel there are three "c" Is this correct? please revise, as the value is much higher that in the other "c". Also describe in the figure legend the statistical analisys and the meaning of the different letters. The same for the other figures.
Figure 7: Please include a statistical analisys similar to the one present in the other figures, and explain it.
Author Response
Reviewer: 3
Comments to the Author:
The paper is interesting and provides interesting information for a broad audience, but there are some points that must be adressed befor beig suitable for publication.
- Introduction: In general aluminum is a problem, specifically in acidic soils which are very common in tropical agriculture. Wouldn't it be easier to acidify the soil that supplementing a toxic like aluminum? Adding aluminum would not be a problem in the long term? Please comment on this in the introduction.
Response: It is true that aluminum as an important problem in acidic soils in tropical agriculture, and the tea plantations are also acidic soils. However, due to the special nutritional characteristics of aluminum in tea plants, aluminum toxicity in tea plantations is not an important issue. Previous studies have suggested that aluminum promotes the root growth of tea plants, and tea plants be considered as aluminum-enriched plants (Shu, W et al. 2003. Chemosphere 52, 1475–1482; Morita A et al. 2004. Phytochemistry 65, 2775–2780.). We carefully revised the content of "Adding aluminum" according to the reviewers' suggestion (Line 95-102). We also added more detailed background of aluminum in tea plantations in the introduction (Line 60-62).
- Figure 2: in the right panel there are three "c" Is this correct? please revise, as the value is much higher that in the other "c". Also describe in the figure legend the statistical analisys and the meaning of the different letters. The same for the other figures.
Response: Figures were checked and redraw according to the reviewer's suggestions. (Line 144)
- Figure 7: Please include a statistical analisys similar to the one present in the other figures, and explain it.
Response: We checked and redrawn the figure based on the reviewers' suggestions. (line 338)
Reviewer 4 Report
Comments and Suggestions for Authors
I propose incorporating hypothesis testing before stating the aim of the paper. This approach would enhance the logical flow of the manuscript, providing a clearer context for the subsequent research objectives.
In line 302, the term "repeated test of two factors (Table 1)" appears ambiguous, suggesting a repeated measures ANOVA which might not be suitable for the data. Additionally, Table 1 seems to be missing. It would be beneficial to clarify the term "repeated test" and ensure the inclusion of Table 1. Furthermore, the addition of a methodological/experimental graphical abstract would enhance the overall understanding of the study.
The inclusion of supplementary materials raises concerns. If supplementary materials are included, they should be properly cited within the main body of the paper. Additionally, all supplementary materials should be adequately described in English to ensure comprehension.
In line 364, considering that the aim of the paper was to "monitor the effects of calcium-aluminium interactions," it is noted that the use of Student's t-test and one-way ANOVA may not be suitable for exploring interactions between factors. Instead, a two-way ANOVA, accounting for the main factors and their interaction, should be conducted, provided that the necessary assumptions of normal distribution and homogeneity of variances are met.
While examining Figures 1 and 2 and marking homogeneous groups, I find a certain inconsistency with the statistical calculations described in the methodology section. Additionally, a note on the visualization of the figures should be addressed, particularly regarding areas where the font size is too small, there are typos, and abbreviations need clarification. Explanations for what the "whiskers" represent on the graphs should be provided. In lines 105-107, this sentence is not supported by data (Figure 1). Generally, the entire results section should be reviewed and reverified for the accuracy of its description. I strongly recommend merging the results and discussion sections.
Author Response
Reviewer: 4
1. I propose incorporating hypothesis testing before stating the aim of the paper. This approach would enhance the logical flow of the manuscript, providing a clearer context for the subsequent research objectives.
Response: Thanks for your constructive comments. This point be added. (Line 101-102)
- In line 302, the term "repeated test of two factors (Table 1)" appears ambiguous, suggesting a repeated measures ANOVA which might not be suitable for the data. Additionally, Table 1 seems to be missing. It would be beneficial to clarify the term "repeated test" and ensure the inclusion of Table 1. Furthermore, the addition of a methodological/experimental graphical abstract would enhance the overall understanding of the study.
Response: Thanks for your constructive comments, we have checked and rewrite the Materials and Methods section (Line 401-415)
- The inclusion of supplementary materials raises concerns. If supplementary materials are included, they should be properly cited within the main body of the paper. Additionally, all supplementary materials should be adequately described in English to ensure comprehension.
Response: Supplementary materials have been checked, modified and cited (Line 144, 170, 338).
- In line 364, considering that the aim of the paper was to "monitor the effects of calcium-aluminium interactions," it is noted that the use of Student's t-test and one-way ANOVA may not be suitable for exploring interactions between factors. Instead, a two-way ANOVA, accounting for the main factors and their interaction, should be conducted, provided that the necessary assumptions of normal distribution and homogeneity of variances are met.
Response: We fully agree with the reviewers' comments. A two-way ANOVA have be conducted and the data be added in tables S1and S2. To focus our works on comparing the effects of aluminum under different calcium conditions, we did not emphasize the interaction effects in the revision.
- While examining Figures 1 and 2 and marking homogeneous groups, I find a certain inconsistency with the statistical calculations described in the methodology section. Additionally, a note on the visualization of the figures should be addressed, particularly regarding areas where the font size is too small, there are typos, and abbreviations need clarification. Explanations for what the "whiskers" represent on the graphs should be provided. In lines 105-107, this sentence is not supported by data (Figure 1). Generally, the entire results section should be reviewed and reverified for the accuracy of its description. I strongly recommend merging the results and discussion sections.
Response: We have rewrite the Materials and Methods section. And redraw the figures. We also merged the results and discussion sections as suggested.
Round 2
Reviewer 1 Report
Comments and Suggestions for Authors
All the suggested changes have been incorporated into the revised manuscript. Two supplementary tables have been added giving the absolute values, as requested, while the authors respond point by point. The paper can be accepted in its present form.
Author Response
- All the suggested changes have been incorporated into the revised manuscript. Two supplementary tables have been added giving the absolute values, as requested, while the authors respond point by point. The paper can be accepted in its present form.
Response: Thanks to the reviewers and editors for their help in revising and publishing the article.
Reviewer 4 Report
Comments and Suggestions for Authors
The manuscript has been improved compared to the previous version; however, regarding a few points, I believe that it still requires more attention from the authors:
-
The research hypothesis should be rewritten. The research hypothesis should take the form of a statement (not a question or guess) and should explain what you expect. Although the purpose of the research was better justified, and these changes should be kept, the research hypothesis is still not clearly formulated in its present form. Please correct this.
-
The suggestion about methodological/experimental graphical abstract was not addressed by the authors.
-
The significant differences should be indicated in the table.
Author Response
- The research hypothesis should be rewritten. The research hypothesis should take the form of a statement (not a question or guess) and should explain what you expect. Although the purpose of the research was better justified, and these changes should be kept, the research hypothesis is still not clearly formulated in its present form. Please correct this.
Response: Thanks for your constructive comments. We add declarative hypotheses. (Line 95-98)
- The suggestion about methodological/experimental graphical abstract was not addressed by the authors.
Response: We added the methodology and experimental graphical abstract as reviewers’ suggestion. (Line 293, 294)
- The significant differences should be indicated in the table.
Response: Significant differences have been added to the tables as suggested (Tables S1, S2).
Round 3
Reviewer 4 Report
Comments and Suggestions for Authors
The paper was improved and can be published in IJMS.